# Application of a ImageJ-Based Method to Measure Blood Flow in Adult Zebrafish and Its Applications for Toxicological and Pharmacological Assessments

**DOI:** 10.3390/biology14010051

**Published:** 2025-01-10

**Authors:** Ferry Saputra, Tzu-Ming Tseng, Franelyne P. Casuga, Yu-Heng Lai, Chih-Hsin Hung, Chung-Der Hsiao

**Affiliations:** 1Department of Chemistry, Chung Yuan Christian University, Taoyuan 320314, Taiwan; ferrysaputratj@gmail.com; 2Department of Orthopaedics, E-Da Hospital/E-Da Dachang Hospital, Kaohsiung 82445, Taiwan; 3School of Medicine for International Students, College of Medicine, I-Shou University, Kaohsiung 82445, Taiwan; ed105483@edah.org.tw; 4Department of Bioscience Technology, Chung Yuan Christian University, Taoyuan 320314, Taiwan; 5Institute of Biotechnology and Chemical Engineering, I-Shou University, Kaohsiung 84001, Taiwan; 6Department of Pharmacy, Research Center for the Natural and Applied Science, University of Santo Tomas, Manila 1008, Philippines; fpcasuga@ust.edu.ph; 7Department of Chemistry, Chinese Culture University, Taipei 11114, Taiwan; lyh21@ulive.pccu.edu.tw; 8Research Center for Aquatic Toxicology and Pharmacology, Chung Yuan Christian University, Taoyuan 320314, Taiwan

**Keywords:** blood flow, ImageJ, Trackmate, zebrafish

## Abstract

Many studies have been developed to calculate the blood flow velocity in adult zebrafish. However, the current method requires high-cost equipment a lot of training before it can be replicated. In this study, we provide a simple, cheap, and non-invasive method to enable the measurement of zebrafish blood flow velocity. By applying the same method as the one previously developed for larvae zebrafish, blood flow velocity can be measured and its sensitivity validated by performing an experiment under several ambient temperature conditions, as well as the presence of a toxicant. The applicability of this method was also checked by applying the same method to other fish. Overall, this method provides an easy, cheap, and robust method that can be used by low- to medium-scale laboratories to contribute to knowledge in the relevant fields.

## 1. Introduction

The heart and other various types of vasculatures, such as veins, arteries, and small capillaries, make up the major parts of the cardiovascular system [1]. The cardiovascular system supplies blood throughout the different parts of the body to provide nourishment. Furthermore, as part of the system, the circulatory vessels can remove waste such as oxygen-poor blood from the body [2,3,4]. In response to different stimuli, the heart’s main function is to regulate the rate of blood flow, so it can control how much blood is transported to various organs of the body through the vascular system [5,6,7]. Intricate coordination between the heart and arteries allows for sufficient blood flow to all bodily components. The delivery of blood to the different organ systems is the primary role of the circulatory system, whereas blood flow occurs because of the pressure caused by the pumping of the heart [8]. In addition, blood flow serves as the most important biologic parameter that can dictate the function of the heart [9,10,11,12].

For cardiovascular physiology, the blood flow endpoint, which is often expressed as the speed or velocity of blood flow, blood volume, or blood pressure, is of utmost significance [13,14,15]. The proper delivery of oxygen and nutrients to all the cells of the body to maintain normal physiological processes depends on adequate blood flow. The removal of metabolic waste products, including carbon dioxide, lactic acid, and other toxins produced during cellular processes, is also enabled by blood flow. Impaired blood flow can cause waste products to build up and damage or malfunction tissue [16]. Blood flow also helps in body temperature regulation [17]. It transports nutrients and immune cells to wounded areas, aiding in healing and preventing infections [18]. Atherosclerosis, coronary artery disease, heart failure, and peripheral artery disease, or PAD, are just a few of the cardiovascular conditions that are characterized by impaired blood flow [19]. Blood flow monitoring can aid in the diagnosis and management of various disorders [20]. In summary, blood flow is a key measurement in cardiovascular physiology because it has a direct impact on how quickly oxygen, nutrients, and chemical signals are delivered to tissues and organs. It is crucial to monitor blood flow and keep it at its ideal level for general wellness, for normal daily activities, and the prevention of multiple heart-related conditions.

Unlike humans, which have a four-chamber double-looped close circulatory system, other animals have developed other circulatory methods in order to adapt to their environment. One example is amphibians, which have developed three-chamber double-looped close circulatory systems for optimal oxygen uptake inside and outside water [21]. Another adaptation can be seen in most fish species, which have two-chamber single-loop circulatory systems to tolerate low oxygen concentrations in water [22]. Although they have a different system, it has a similar function to the human homolog; both have hearts to pump blood to the whole body, which is regulated by the heart muscle, and blood vessels, which enable blood to flow through the whole body [23,24]. Thus, learning animal cardiovascular dynamics can also help us to understand the human cardiovascular system.

As a diagnostic tool, blood flow measurements are very important in assessing the functions of the heart, as well as in the diagnosis of several cardiac disorders. Previous studies have been carried out to calculate blood flow velocity in animals (Table 1). To measure blood flow, a variety of traditional procedures, from non-invasive to more intrusive, are used. Doppler ultrasound [25], ultrasound imaging velocimetry (UIV) [26], and vector Doppler [27], are some of the methods that have been investigated for measuring non-invasive blood flow velocity profiles. Recent developments in high-frame-rate imaging technology offer new possibilities for flow estimation [28]. The analysis of high-speed video recordings of fish blood vessels can be used in estimating the speed of blood flow by closely monitoring the course of movement of the red blood cells of zebrafish [29]. It is crucial to remember that every technique has unique benefits and drawbacks, including sensitivity, invasiveness, and compatibility for various species and sizes. The precise study objectives, the size of the animal, the target blood arteries, and the accessible tools and knowledge are frequently taken into consideration while selecting a procedure to minimize the usage of animals, following the reduce, reuse, and recycle policy. Following the directions of several countries, procedures that cause stress or harm to the animal, like handling before, during, and after the experiment, also need to be reduced and taken into consideration when evaluating the usability of the method [30,31].

For studying the cellular and molecular principles underpinning heart development and regeneration, zebrafish are regarded as the perfect model organism. The zebrafish’s history as a model organism for cardiovascular research has been surprisingly short and quite successful. Zebrafish have a closed circulatory system and a very similar cardiac cycle to that of humans [40]. They are small in size, easy to grow, and have transparent body at the larvae stage that makes it easy to observe the function and development of the cardiovascular system [41]. While their genes share 70% homolog with humans, zebrafish are also sensitive when they are exposed to chemicals that can cause changes to the cardiovascular system, and they display a similar toxicity profile to that of humans [42,43,44,45,46,47]. Apart from that, a lot of information is already available regarding their genetic mutations and many methods have been developed to perform gene manipulation in zebrafish [48,49,50]. Due to this trait, zebrafish is a good animal model for cardiotoxicity studies.

In most zebrafish studies, cardiac function and blood flow hemodynamics of the animals must be measured to indicate the effects of the interference on the cardiovascular system, because zebrafish experimentation typically interferes with cardiac functions or structure [51,52]. However, up to now, the method for calculating blood flow velocity in adult zebrafish has had major drawbacks; the high cost of the required equipment and the need for a professional to perform it. In this study, our aim is to develop a simple, cheap, and sensitive method to check the vascular system of adult zebrafish. By taking advantage of the transparency in the zebrafish tail, measurements of the blood flow velocity can be performed in zebrafish by applying the method previously developed for larvae zebrafish [29]. After that, we measured the sensitivity of zebrafish to different ambient temperatures and chemicals that cause cardiovascular imbalance, namely fenproparthrin and ponatinib, in order to validate the sensitivity of the parameter and make sure that the parameter can accurately represent the vascular system. Furthermore, after validation, zebrafish were exposed to copper oxide combined with carbofuran to check for possible vascular alteration after exposure to pollutants. In the end, the versatility of this method was also explored by reproducing the same method with other fish. By developing this method, we created a possibility for small- to middle-sized laboratories to use this method and to also contribute to and hasten the advancement in this field.

## 2. Materials and Methods

### 2.1. Animal Husbandry

In this study, pet store-purchased (PET) zebrafish, tiger barb (*Puntigrus tetrazona*), Convict cichlid (*Amatitlania nigrofasciata*), black tetra (*Gymnocorymbus ternetzi*), and medaka (*Oryzias latipes* and *Oryzias woworae*) were purchased from a local fish supplier (Zhongli District, Taoyuan, Taiwan). Before use, all the fish were kept in the zebrafish facility for one month for acclimatization and synchronize other growth condition parameters that might affect the experiment. All the fish were kept in a tank with circulating water at ~26 °C, with a pH of 6.5–7.5 and a 10/14 h dark/light cycle. The NH_3_ concentration was kept under 2 ppm, while the NO_3_ concentration was kept under 40 ppm, with water conductivity being maintained at 300–1500 μS. The water was filtered using a UV filter to ensure the quality of the circulating water. All the fish were fed twice a day with brine shrimp and dry food for balanced nutrients. All the regular care, routine culture, and maintenance of zebrafish were carried out according to the published procedure by Avdesh et al. [53]. All tests were carried out in accordance with the rules approved by the Institutional Animal Care and Use Committees (IACUCs) of the Chung Yuan Christian University (approval number 110016). Furthermore, all the fish handling and methodologies used also followed the guidelines of the European Parliamentary Council regarding the protection of animals used for scientific purposes [30].

### 2.2. Chemical Preparation

Fenpropathrin (FEN) and ponatinib (PON) compounds with >95% purity were purchased from Aladdin Bio-Chem Technology Co., Ltd. Shanghai, China. Copper oxide nanoparticle (20% *w*/*v*) was purchased from Hangzhou Zhiti Purification Technology Co., Ltd., Hangzhou, China. Carbofuran was purchased from Sigma-Aldrich St. Louis, Missouri, United States and diluted with dimethyl sulfoxide (DMSO) to concentration of 100 mg/L. FEN was diluted with acetone to the concentration of 10 g/L as the stock solution. It was further diluted with ddH_2_O at the time of exposure. PON was diluted with DMSO to a concentration of 20 g/L and later diluted with ddH_2_O at the time of exposure.

### 2.3. Blood Flow Velocity Validation

For validation of the sensitivity of the blood flow velocity in the tail fin, the PET zebrafish were exposed to various ambient temperatures. A day before measurements were recorded, the fish were placed into incubators at temperatures of 15, 20, 26, and 32 °C, to make sure that any changes in zebrafish blood flow velocity were not caused by the heat shock after a sudden change in temperature. To maintain the oxygen concentration needed by the fish, an aerator was placed in the fish tank for the whole incubation period. Furthermore, two blood vessel positions at the start of the tail fin and the blood vessel position near the end of the tail fin, parallel to the heart position, were checked to obtain the best possible position for blood flow velocity measurements.

In the validation step, zebrafish were exposed to fenpropathrin (FEN) to check whether the blood flow velocity in the tail fin was sensitive to changes caused by a compound that can increase the aortic blood flow velocity. Following a previous study on larvae, the zebrafish were orally exposed to FEN for 24 h at concentrations of 0.1, 0.01, and 0.001 mg/kg body weight (*w*/*w*). In addition, acetone at a concentration of 0.01% was used as the positive control. Similarly to FEN, PON was also provided orally at concentrations of 0.1, 0.01, and 0.001 mg/kg body weight (*w*/*w*), and 0.1% DMSO was used as the positive control [43,54]. The oral exposure was carried out by gently placing the fish into 0.1% tricaine methanesulfonate (MS-222) solution until they reached stage III anesthesia, shows by a total loss of equilibrium and less movement of operculum plate [55,56]. After that, the fish were placed on a soft sponge and the desired volume of compound was slowly inserted into the oral gavage using a soft tip, without damaging the inside of oral gavage. The fish were then immediately placed into an aerated water system to make sure they could wake up completely without any problems. In the experiment, only zebrafish weighing between 0.4 and 0.8 g were used, to decrease the possibility of variation being caused by body weight.

### 2.4. Power Analysis for Blood Vessel Selection

The selection of an appropriate place for blood flow velocity measurements was carried out using statistical calculation [57]. About twenty zebrafish were selected and the blood flow velocity at both selected locations was measured. After that, a power analysis was performed to obtain the necessary sample size (*n*) number in both locations, and the location with the lowest necessary *n* number was selected as the appropriate place for further testing. The necessary *n* number was calculated using the following formula:$$n=\frac{Z_{\alpha /2}^{2}\times \sigma^{2}}{E^{2}}$$

In the formula, *n* is equal to the required sample number, $Z_{\alpha /2}^{2}$ is the critical value of the standard normal distribution corresponding to the desired level of confidence (1 − alpha/2), σ is the standard deviation, and E is the desired margin of error. For the selection, the desired level of confidence was set to a minimum of 80% and the desired margin of error was set to 5% of the mean value [58].

### 2.5. High-Speed Videography

The high-speed videography was carried out using a high-speed charged-coupled device (CCD) camera (AZ Instrument, Taichung City, Taiwan), mounted onto an inverted microscope (ICX41, Sunny Optical Technology, Yuyao, Zhejiang, China), which was capable of recording at 200 frames per second (fps). A Hoffman modulation objective lens with 40× magnification was used to enhance the quality of the blood cell visualization. To record the blood flow velocity, the fish was first anesthetized by gently placing them into 0.1% MS-222 solution, in the same way as for the oral exposure experiment. After that, the fish was placed gently into a 5 cm Petri dish, with its face facing left, thereby showing its lateral left side. About 200 mL of ddH_2_O was added to the tail fin before measurements, to make sure that the tail fin expanded to its maximum size. The measurements were conducted for 10 s on the tail fin of the zebrafish. As soon as the measurements were finished, the fish was moved to well-aerated water to make sure the fish woke up from the deep anesthesia stage without any problems.

### 2.6. Blood Flow Velocity Measurements

Blood flow velocity measurements were carried out by adapting the previously developed method by Santoso et al. [29]. Based on a pre-testing experiment, zebrafish larvae, at 3 days post-fertilization, were found to have a similar blood cell size to adult zebrafish (Appendix A). Thus, a setting was used in larvae blood flow calculation which is also applicable in adult zebrafish. The analysis of the blood flow velocity measurements was conducted using the Trackmate plug-in on ImageJ software version 1.52v under FIJI distribution (https://imagej.net/software/fiji/ accessed on 22 August 2023) [59]. The video was first converted into an ImageJ-friendly format using VirtualDub2 software (http://virtualdub2.com/ accessed on 22 August 2023). After conversion, the video was uploaded to imageJ and the Trackmate program was used to track the blood flow velocity. The Trackmate plug-in reported every blood cell detected in the video, and the Trackmate results were further processed using the authors’ originally developed virtual basic application (VBA) macro in Excel, which allowed for batch analysis for several raw data points from Trackmate at once.

### 2.7. Blood Flow Velocity Measurements After Waterborne Exposure to Carbofuran and Copper Oxide

After validation, the newly developed method was used to measure blood flow velocity following waterborne exposure to carbofuran (CAR) and copper oxide (CuO). Pretesting using the same method was performed on both compounds, and a concentration of 0.1 ppm was selected for CAR and 100 ppm for CuO. The zebrafish were exposed to both compounds in combination, and the blood flow velocity was checked after 24 h of exposure. About ten zebrafish were placed into a 3 L glass beaker, and 2 L of the compound was added, with the addition of an aerator in the beaker glass to ensure airflow inside. The whole overview of the experimental design can be seen in Figure 1.

### 2.8. Statistical Analysis

All statistical analyses were carried out using GraphPad Prism 8 software (GraphPad Inc., La Jolla, CA, USA). The experiment was conducted in triplicate and all the data were pooled together to be analyzed with appropriate statistical tools based on the normality and standard deviation of the data. To check the significant difference between the velocity in the main blood vessel and in the blood vessel at the end of the tail fin, an unpaired *t*-test was used, with *p* < 0.05. To analyze the significant differences between different temperatures and age groups, an ordinary one-way ANOVA, followed by a Dunnet’s multiple comparison test, was performed.

## 3. Results

### 3.1. Blood Flow Velocity Measurements on the Blood Vessels at the Base of the Tail Fin and the Tip of the Tail Fin

Blood flow velocity measurements were carried out in two parts of the tail fin, namely the base and tip of the zebrafish tail fin. Taking measurements on these parts has advantages and disadvantages. When recording was conducted on the tail fin base, less branching of the blood vessel was observed compared to the tail fin tip, which resulted in better accuracy when analyzing the video. However, it must be noted that, although the tail fin base had less blood vessel branching, it resulted in less visibility as more pigment was concentrated around the tail fin base; sometimes, this disturbed the observation if it was positioned exactly at the blood vessel (Figure 2A,B). In terms of velocity, the blood cells in the tail fin base had a significantly higher velocity compared to the tail fin tip for both maximum (1141 ± 214.3 and 822 ± 255.5 µm/s) and average blood flow velocity (524.9 ± 45 and 461.3 ± 128.2 µm/s) (Figure 2C,D). Example videos of the blood vessels in the base and tip of the tail fin can be found as Appendix A, respectively.

As previously mentioned, the measurement recording position was selected using a power analysis of blood flow velocity from both locations. By following the above-mentioned formula, the ideal necessary sample size for maximum blood flow velocity at the base and tip of the tail fin was 23.17, which was rounded to 24, and 63.47, which was rounded to 64, respectively. For the average blood flow velocity, 4.83, which was rounded to 5, and 50.77, which was rounded to 51, were calculated as the ideally necessary sample sizes for the tail fin base and tail fin tip, respectively. Based on the power statistic and the other factors observed, further measurement recordings focused on the base of the tail fin.

### 3.2. Blood Flow Velocity at Different Ambient Temperatures

The next step was to validate whether the blood flow velocity in the tail fin base was sensitive enough to changes in the environmental conditions, which, in this case, were different ambient temperatures. From the results, it could be observed that the maximum blood flow velocity was significantly lower at 15 °C (807 ± 290.6 µm/s) compared to the velocity measured at 32, 26, and 20 °C (1246 ± 330, 1198 ± 245.5 and 1257 ± 377.3 µm/s). For the average blood flow velocity, a similar result was observed; at 15 °C (401.3 ± 110.7 µm/s), the blood flow velocity was significantly lower compared to other temperatures. Furthermore, it was observed that the average blood flow velocities at 32 (556.1 ± 109.9 µm/s) and 20 °C (484.3 ± 90.78 µm/s) were significantly different. However, if compared with the velocity at 26 °C (525.2 ± 54.05 µm/s), both temperatures showed no significant difference (Figure 3B). The exact *p* value for the one-way ANOVA can be found in Appendix A. Based on these data, we confirmed that the blood flow velocity in the tail was sensitive to changes in the ambient temperature.

### 3.3. Sensitivity of Blood Flow Velocity to Fenpropathrin and Ponatinib

Another validation was carried out to check the sensitivity of zebrafish blood flow velocity to chemicals. FEN (fenpropathrin), a pyrethroids insecticide, has been previously shown to increase the blood flow velocity in zebrafish larvae after acute exposure and 24 h incubation [54]. Similarly, a significant increase was observed in adult zebrafish blood flow velocity 24 h after the oral administration of FEN. It was observed that FEN caused a dose-dependent increase in maximum blood flow velocity starting from 0.01 mg/kg exposure (1029 ± 253.7 µm/s, *p* = 0.272) (Figure 4A). Furthermore, a significant increase was also observed in the average blood flow velocities of the 0.1 (499.7 ± 94.09 µm/s, *p* = 0.0207) and 1 (499.8 ± 94.09 µm/s, *p* = 0.0276) mg/kg groups (Figure 4B), which means that exposure to FEN can cause a significant increase in blood flow velocity; therefore, blood flow velocity is sensitive to this chemical.

PON (ponatinib), an anticancer agent, was selected for further validation of the parameter, since, previously, it has been proven to cause some damage to the cardiovascular system and significantly decrease blood flow velocity in zebrafish larvae [43]. In line with the previous study, the current findings of this investigation showed that the maximum blood flow velocity significantly increased in both the 0.1 (842.9 ± 298.4 µm/s, *p* = 0.0417) and 1 (840.2 ± 309.6 µm/s, *p* = 0.0354) mg/kg body weight groups, compared to the control group (Figure 5A). Similarly, a significant decrease in the average blood flow velocity was also observed in both the 0.1 (405.9± 102.3 µm/s, *p* = 0.0004) and 1 (395.8 ± 119.6 µm/s, *p* = 0.0001) mg/kg groups (Figure 5B). The data regarding FEN and PON supports the conclusion that the blood flow velocity observed in the base of the tail fin may be a good parameter to estimate the condition of the vascular system of zebrafish.

### 3.4. Blood Flow Velocity in Copper Oxide- and Carbofuran-Exposed Zebrafish

Zebrafish exposure to CuO (copper oxide) and CAR (carbofuran) was conducted to check the possibility of a synergistic effect on blood flow velocity. CuO is one of the pollutants that can be found in water bodies. Together with CAR, an emerging persistent pollutant, it has been shown to cause a synergistic effect and to alter the cardiovascular systems of zebrafish larvae [44]. Water immersion exposure was conducted for 24 h at sub-effective concentrations (100 ppm of CuO and 0.1 ppm of CAR), following the methodology of a previous study, and pretesting was conducted to check the possibility of a synergistic effect of both compounds (Appendix A). However, no significant difference was observed after co-exposure to both compounds (Figure 6).

### 3.5. Blood Flow Velocity Measurements in Other Fish Species

Aside from zebrafish, several other fish species were tested to assess the versatility of this method, including black tetra, tiger barb, Convict cichlid, and medaka fish. The same protocol was used for all the fish, and the measurements were also taken in the same section of the tail fin. We observed that black tetra have similar maximum and average blood flow velocities to zebrafish (Figure 7A). However, both medaka species have significantly lower maximum and average blood flow velocities than both zebrafish and black tetra (*p* < 0.0001). Despite this, no significant difference was observed between the two medaka species. A calculation of the required sample size was also carried out for other fish species, and it showed that black tetra needed at least 20 samples, while *Oryzias woworae* and *Oryzias latipes* needed 62 and 171 samples, respectively.

Our observations also showed that, due to the heavy pigmentation surrounding the tail fin base, visualizing the blood vessel proved to be difficult in tiger barb, as vessel visualization is critical for this method (Figure 7B). A similar issue arose with Convict cichlid, where the blood vessel’s position near the edge of the bone made clear visualization difficult. Furthermore, in Convict cichlid, both the arterial and venous vessels in the fin tail are positioned overlapping each other, making accurate measurements difficult (Figure 7C).

## 4. Discussion

Although there are a lot of studies that attempt to measure blood flow velocity in adult zebrafish, the authors believe that a small- to medium-scale laboratory would encounter some problems replicating the method, due to its limitations regarding equipment cost. In this study, the authors attempted to measure the blood flow velocity in the tail fins of zebrafish using a simple setup, namely a microscope mounted on a high-speed CCD camera, in order to achieve a good visualization of the blood vessels around the tail fin area. This method was adapted from a previous study that used a simple microscope–camera setup to calculate the blood flow velocity in zebrafish larvae [29]. The authors also discovered that this parameter was also sensitive to changes caused by different ambient temperatures and by some compounds that alter the cardiovascular system, namely FEN and PON. Furthermore, this method was further validated by exposing zebrafish to combined CuO and CAR, which, contrary to the expected results, did not result in any statistically significant differences in blood flow velocity after exposure. The method’s versatility was also challenged by performing the experiment on several other fish, which showed that this method also works for other fish that have a similar tail fin structure.

In this study, the measurement of blood flow velocity was carried out on one of the parts of the tail fin of the zebrafish. Although the visualization of the blood vessels at the dorsal and anal fins is possible in adult zebrafish, the authors believe that visualization at the tail fin will give a better representation of the cardiovascular system in zebrafish, as the blood vessels in the tail fin are parallel to the dorsal aorta [60,61]. The results confirmed that the velocity at the base of the tail fin was more prominent and faster compared to the velocity at the tip of the tail fin. This makes sense because base tail fin position was closer to the dorsal aorta than that at the tip of the tail fin; the blood vessel in the tip of the tail fin is actually a branch of the blood vessel in the base of the tail fin [62,63,64]. It is a well-known concept that, as the total cross-sectional area of the blood vessel at the capillaries increases, the velocity of the blood flow will decrease [65,66]. Furthermore, the farther the blood vessel is from the heart, the more the resistance of vessel walls will increase, which affects the velocity of the blood flow. This is a natural phenomenon, as the exchange of nutrients and waste from metabolism carried out in capillaries requires a slower blood flow for the process to take place correctly [67,68].

Significant differences in blood flow velocity at different ambient temperatures were also observed in this study. It is a well-known theory that hot environmental conditions will increase blood flow velocity, while cold environmental conditions decrease it. In line with this, the zebrafish tested in this study tended to have an increased blood flow velocity at high temperatures and a slower velocity at low temperatures. This natural phenomenon helps to regulate the core internal temperature and is called thermoregulation. The body will expand blood vessels to increase blood circulation in the body to help release heat, and vice versa; these are called vasodilation and vasoconstriction, respectively [69,70]. Disruption to thermoregulation could cause hypothermia or hyperthermia; that is, if the condition is prolonged enough, it can lead to organ failure, coma, or even death [71,72]. In fish, as well as disrupting the cardiovascular system, disruption to thermoregulation will also cause some changes in swimming behavior [73,74].

The validation in this study showed that the oral exposure of zebrafish to FEN caused a significant increase in blood flow velocity in a dose-dependent manner, even after 24 h. FEN, a pyrethroids pesticide, is designed to affect the voltage-gated sodium channel by disturbing depolarization and causing an influx of sodium ions [75]. As Type I pyrethroids, exposure to FEN causes restlessness, incoordination, hyperactivity, prostration, and paralysis [76]. Aside from the sodium channel, FEN also acts by disrupting the calcium and chloride channels [77], and an in silico study also confirmed these activities by showing the high binding affinity of pyrethroids to binding sites that cause such effects [54]. Previous reports on human exposure show that FEN can cause Parkinson’s disease after 6 months of oral exposure, due to the degeneration of dopaminergic neurons in the brain [78,79]. In zebrafish, the degradation of neurons and alterations in locomotion and social behavior were observed, together with a significant increase in heart size and an increase in heart rate, following FEN exposure [54,80,81]. In a previous study, FEN also caused a significant increase in blood flow velocity in zebrafish, which was also observed in this study.

Another validation, conducted with the use of PON administered orally to zebrafish, showed that blood flow velocity in the tail fin significantly decreased after 24 h exposure. PON, a third generation anti-leukemia agent, works by binding to various tyrosine kinase proteins and inhibiting their action [82]. It has been shown to possibly even help with chronic myeloid leukemia, which causes the T315L mutation that has been proven to be resistant to previous generations of anti-leukemia drugs, such as imatinib. Although PON demonstrated good results for treating patients with leukemia, it also exhibited severe side effects, including cardiotoxicity; a recent study reported that it caused myocardial and systemic inflammation due to the activation of the S100A8/A9-TLR4-NLPR3-IL-1β signaling pathway [83]. Due to this trait, PON become a second-in-line drug that will be used whenever other possible drugs fail to treat leukemia. Previous studies show that PON can cause severe cardiovascular problems in zebrafish. Zebrafish larvae exposed to PON had significantly decreased stroke volume, ejection fraction, and cardiac output, followed by a significant decrease in heart rate. A significant decrease in blood flow velocity has also been observed, together with vasoconstriction in the dorsal aorta, after PON exposure, showing that it is a potent cardiotoxic agent [43,84]. Based on both validations, it was observed in this study that blood flow velocity in the zebrafish tail fin was also affected by chemicals that affect the zebrafish cardiovascular system.

CuO is an important compound used in many industries as a catalyst, a pigment, a sensor, a preservative, and various other applications. However, due to excessive use, it can be easily found in bodies of water as well as soils, which may become a growing concern in the future. Previous studies have shown that CuO can be found in aquatic bodies and that CuO toxicity is mainly caused by an imbalance in reactive oxygen concentrations in zebrafish which caused cell death, leading to tissue stress [85,86]. Sun et al. (2016) reported that exposure to a concentration of 12.5 mg/L CuO caused phenotypic defects and inflammatory responses in zebrafish larvae [87]. A chronic 30-day exposure, as reported by Mani et al. (2020), caused muscle toxicity in zebrafish in a dose-dependent manner, which shows that it is indeed toxic to zebrafish [88].

On the other hand, carbofuran, a carbamate pesticide, has long been used to control insects and nematodes in agricultural fields, leading to carbofuran contamination in bodies of water [89,90]. Its toxicity is manifested through the reversible inhibition of acetylcholinesterase activity; therefore, it is extremely toxic to mammals and aquatic animals [91,92,93,94]. It was also reported that carbofuran causes anxiety-like behaviors in female zebrafish [95]. It also shortened the lifespans of wild-type zebrafish by disturbing cellular autophagy functions and accelerated senescence though the regulation of the nrf2 signaling pathways [96]. Another study also reported that the long-term consumption of feed mixed with carbofuran by Wistar rats for 30 days caused alterations to a drug-metabolizing enzyme in their livers, which shows the potent toxicity of carbofuran [97].

Previously, CuO and carbofuran demonstrated a synergistic effect in zebrafish larvae, adversely affecting cardiovascular function [44]. However, in the present study, no alteration was observed in zebrafish blood flow velocity after exposure. This might be attributed to the mechanism of action of both compounds, which involves the accumulation of metabolites, especially reactive oxygen species (ROS) [98,99]. In particular, since adult zebrafish have more developed and better defense mechanisms against ROS compared to larvae, it might take more time for the compound to cause an effect [100]. Furthermore, compared to the readily absorbed copper sulfate, CuO has a lower absorption rate, but it will persist longer in the system. Thus, 24 h incubation might not be enough for it to cause observable effects. Based on studies, copper absorption is necessary to exhibit CuO effects. However, this could be interesting for future study, especially for toxicity studies, as this method enabled the long-term observation of toxicants’ effects on zebrafish.

The current study showed that other species have different required sample sizes to achieve 80% data confidence, with the medaka species needing 62 and 171 samples for *Oryzias woworae* and *Oryzias latipes,* respectively. This is due to intraspecies and biological variation. For zebrafish, although they were sourced from the local aquarium, some filtering was carried out based on the fish size. However, for medaka, size filtering was not performed due to the limited quantity. Despite this, the experiment was proven successful, as the current method was also able to calculate blood flow velocity in medaka tail fins. Therefore, to use this method on medaka in the future, standardization needs to be conducted to make sure there is less variation, in order to therefore increase the accuracy of the data.

In this study, the calculation of blood flow velocity in tiger barb and Convict cichlid could not be performed. This represents the main limitation of the current method, as the visualization of the blood vessels was critical for this method to function effectively. Pigmentation was the main problem in tiger barb, as the blood vessels were blocked by pigment and they could not be tracked with Trackmate, raising concerns regarding its accuracy. Another challenge was the presence of another blood vessel overlapping with the target vessel, which happened during measurement recordings for Convict cichlid. Since Trackmate calculates all moving particles within the specified region of interest (ROI), regardless of their direction, any overlapping blood vessel, particularly venous vessels that move in completely different directions, were also included in the report; this was hard to differentiate during the downstream process. Moreover, arterial and venous blood vessels have different flow speeds, which further compromised the accuracy of the results, especially in calculating the average blood flow velocity, as all detected blood cells were included in the calculation [29]. However, with advancements in artificial intelligence, especially in image segmentation, this issue can be addressed by applying a filter to enhance the brightness of the region of interest or using an advanced algorithm to separate the flows, e.g., OpenCV, which will be interesting for future study [101].

Although this method was created for simple blood flow velocity tracking in zebrafish larvae, there are several fields that can benefit from applying this method. With the possibility of non-invasive observation, researchers can carry out observations of the same individual at several different time points to observe the long-term effects of a compound on the vascular system of a fish. Furthermore, the current study uses this method on zebrafish tail fins; however, this method is applicable to other animals that have transparent bodies, like glass catfish, transparent shrimps, or even glass frogs, to obtain a better understanding of their cardiovascular systems.

## 5. Conclusions

This study provides a simple and cost-effective solution to carry out the quantification of blood flow velocity in adult zebrafish. Based on a simple setup comprising a high-speed CCD camera and an inverted microscope, measurements of blood flow velocity, which is an important cardiovascular performance parameter, were be obtained from the tail fin base of adult zebrafish. Furthermore, the dose-dependent reduction in the blood flow velocity following incubation at different temperatures showed that this parameter is sensitive to changes in the environmental temperature. The sensitivity was further checked by using two pharmacological drugs, ponatinib and fenpropathrin, that were previously shown to cause significant alterations in blood flow velocity; they showed similar results to previous testing. Another test, using a combination of copper oxide and carbofuran, also showed interesting results; there was no significant change in blood flow velocity following acute 24 h exposure, compared to a previous report that confirmed the alteration of cardiovascular function in zebrafish larvae. The current method was also proven to be versatile, as it can also perform well on other animals, like tiger barb and medaka. As a side note, this method of quantification was heavily dependent on the quality of the video and the presence of pigmentation or other objects directly affecting the clarity of the blood vessel visualization. Overall, the results of this study can help other researchers to carry out testing in adult zebrafish of drugs that target the vascular system, which were previously only tested on larvae.

## Figures and Tables

**Figure 1 biology-14-00051-f001:**
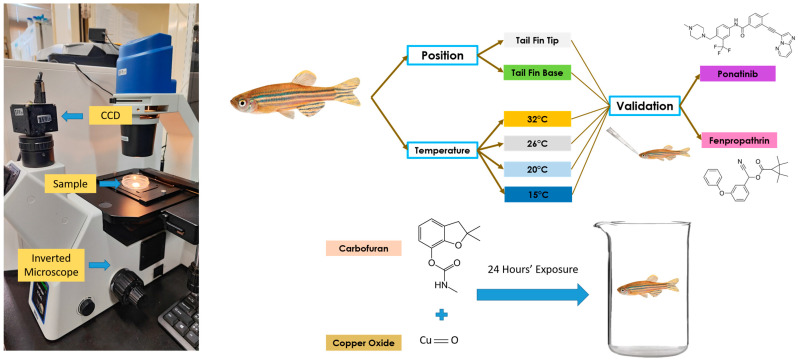
Video recording setup and overview of the whole experiment to validate blood flow velocity measurements in adult zebrafish. The **left panel** shows the microscopic setup used to conduct high-speed videography for blood flow tracking. The **right panel** shows the experimental design for studying position and temperature effects on blood flow measurements, and the potential effect of chemical exposure on blood flow alterations.

**Figure 2 biology-14-00051-f002:**
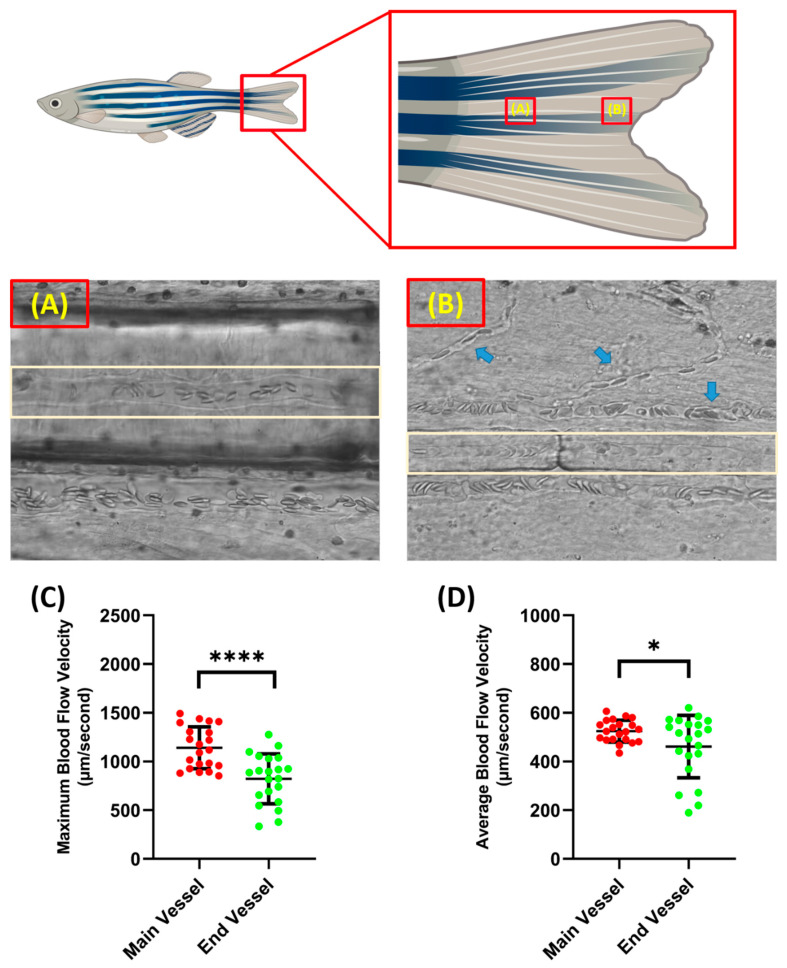
Condition of blood vessel at the base (**A**) and the end of the tail fin (**B**) (light brown color shows the position of the selected blood vessel for measurement and the blue arrow shows the position of the blood vessel branch). Maximum (**C**) and average (**D**) blood flow velocity of zebrafish at different tail fin positions. The statistically significant difference was calculated using *t*-test (*n* = 20, * *p* < 0.05 and **** *p* < 0.0001).

**Figure 3 biology-14-00051-f003:**
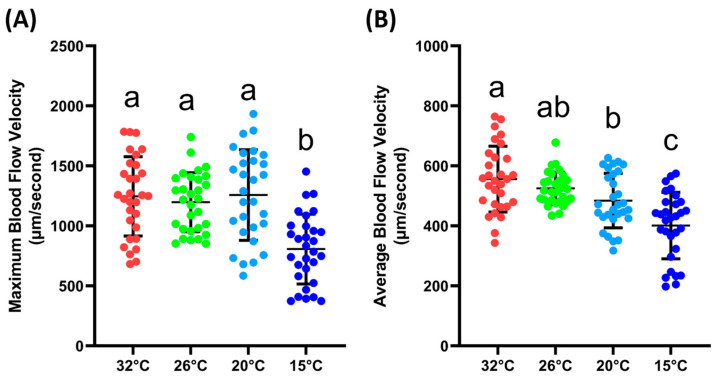
Maximum (**A**) and average (**B**) blood flow velocity of zebrafish at various ambient temperatures. Character a, ab, b, and c above the scatter dots show the significant difference between each group, with the degree of significance adjusted to *p* < 0.05. The statistical difference was calculated using an ordinary one-way ANOVA with Dunnet’s multiple comparison test (*n* 32° = 30, *n* 26° = 30, *n* 20° = 29, *n* 15° = 30).

**Figure 4 biology-14-00051-f004:**
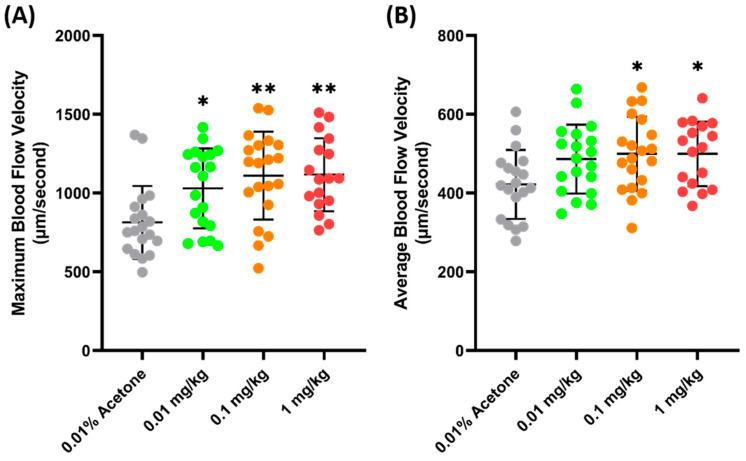
Maximum (**A**) and average (**B**) blood flow velocity of zebrafish after 24 h exposure to FEN. The statistical difference was calculated using an ordinary one-way ANOVA with Dunnet’s multiple comparison test (* *p* < 0.05, ** *p* < 0.01) (*n* Control = 19, *n* 0.01 mg/kg = 19, *n* 0.1 mg/kg = 20, *n* 1 mg/kg = 17).

**Figure 5 biology-14-00051-f005:**
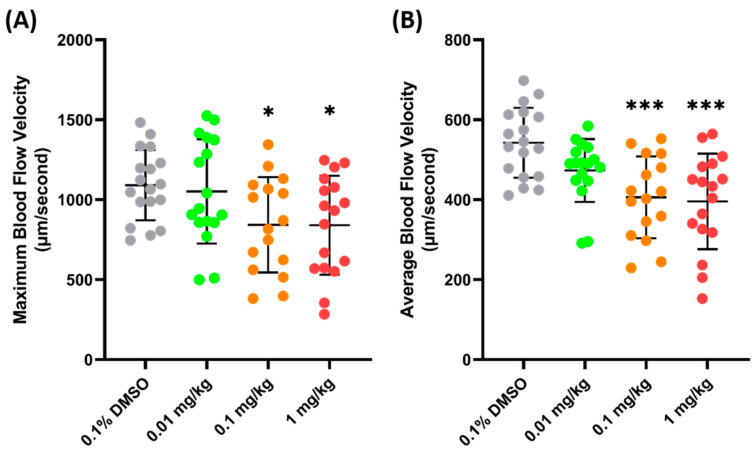
Maximum (**A**) and average (**B**) blood flow velocity of zebrafish after 24 h exposure of PON. The statistical difference was calculated using an ordinary one-way ANOVA with Dunnet’s multiple comparison test (* *p* < 0.05, *** *p* < 0.001) (*n* Control = 18, *n* 0.01 mg/kg = 17, *n* 0.1 mg/kg = 16, *n* 1 mg/kg = 17).

**Figure 6 biology-14-00051-f006:**
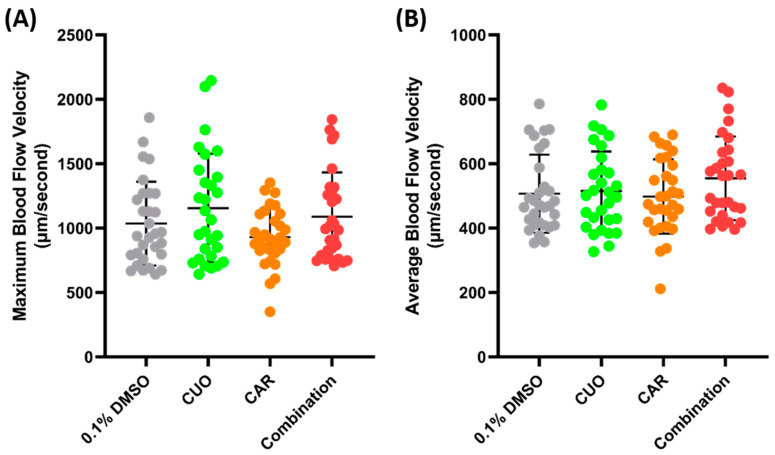
Maximum (**A**) and average (**B**) blood flow velocity of zebrafish after 24 h exposure of copper oxide (CuO) at 100 ppm, carbofuran (CAR) at 0.1 ppm, and a combination of both compounds. The statistical difference was calculated using an ordinary one-way ANOVA with Dunnet’s multiple comparison test, with the degree of significance set to *p* < 0.05 (*n* Control = 30, *n* Copper = 29, *n* Carbofuran = 30, *n* Combination = 29).

**Figure 7 biology-14-00051-f007:**
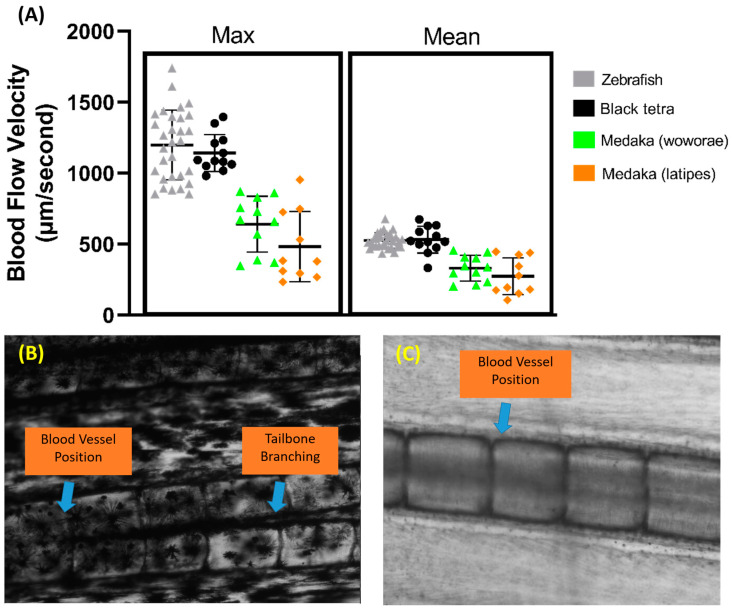
Comparison of maximum and average blood flow velocity of zebrafish, black tetra, *Oryzias woworae*, and *Oryzias latipes* on the tail fin (**A**). The data were presented as mean ± SD and the statistical difference was calculated using an ordinary one-way ANOVA with the degree of significance set to *p* < 0.05 (*n* Zebrafish = 30, *n* Black tetra = 12, *n Oryzias* woworae = 11, *n Oryzias latipes* = 10). The overview of tiger barb (**B**) and Convict cichlid (**C**) caudal fin condition showing the position of blood vessel.

**Table 1 biology-14-00051-t001:** Previous study attempts of blood flow velocity measurements in animals.

Year	Method	Target	References
2008	High-frame rate duplex ultrasound biomicroscopy	Mouse and zebrafish	Xu et al. [32]
2012	Line-scannning particle image velocimetry	Mouse	Kim et al. [33]
2013	Retrospective Doppler-gated echocardiography system	Zebrafish	Liu et al. [34]
2015	High-resolution echocardiography system	Zebrafish	Huang et al. [35]
2016	Laser speckle contrast imaging	Atlantic cod	Ambrus et al. [36]
2019	High-speed video imaging	Zebrafish larvae	Santoso et al. [29]
2020	High-frequency ultrasound deformation	Zebrafish	Chiang et al. [37]
2020	Ultrasound localization microscopy	Mouse	Espindola et al. [38]
2024	Time-resolved laser speckle contrast imaging	Rodent	Faraneh et al. [39]

## Data Availability

We have full control of all primary data and agree to allow the journal to review our data upon request.

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
