# Peer review of "Application of a ImageJ-Based Method to Measure Blood Flow in Adult Zebrafish and Its Applications for Toxicological and Pharmacological Assessments"

_biology, 2025, doi:10.3390/biology14010051_

Round 1
Reviewer 1 Report
Comments and Suggestions for Authors
Please read the whole manuscript carefully and solve the comments and suggestions.

Language quality is good but need to improvement
Author Response
Comments and Suggestions for Authors
Please read the whole manuscript carefully and solve the comments and suggestions. peer-review-42480785.v1.pdf
Comments on the Quality of English Language
Language quality is good but need to improvement
Thank you for the comment and suggestion. The updated manuscript has been updated following the reviewer comment and suggestion. However there are several point that the authors think need to be address:
- Regarding the point of moving the fourth paragraph at the introduction to the method section. The author did not think that it is appropriate to do due to the content of the paragraph. The paragraph mentioned describe the previous attempt to calculate the cardiac performance in animal without any relation to the methodology perform in the current study thus the authors think that it didn’t fit into the material and method section.
- The other thing need to be address is the presence of table in the introduction section. Although the authors agree with the reviewer concern, the authors still think that it is necessary to put the mentioned table as it can drastically reduce the words count and make it easier for the reader to look at the intention of the authors.
- Another point is the suggestion to breakdown the section 2.1 into two sub-section. The authors think that the breakdown into two subsection won’t be need because the suggestion of section “study site” will be too short and it also already address implicitly that the study was done in Taiwan in the first sentence of 2.1 section. However, the authors agree that more information regarding the fish handling especially the growth environment need to be address further. Thus, the updated manuscript has been revised according to the reviewer suggestion.
Reviewer 2 Report
Comments and Suggestions for Authors
-
The introduction addresses two main topics: the importance of blood flow for cardiovascular physiology and methods for its measurement in fish. However, the transition between these topics could be clearer. The relationship between blood flow in humans and its measurement in fish is not explicitly stated, which may hinder the reader's understanding of the overall relevance. It is recommended to reorganize the introduction to better integrate these themes.
-
Many sections describe blood flow in general terms, with statements like "blood flow serves as the most important biologic parameter". While true, such claims would be more impactful if supported by specific examples or data. This approach helps justify the study's relevance and its contribution to the field.
-
References are used to support assertions, but some information could be enriched with more recent or specific citations, especially regarding measurement techniques in fish. The suggested article (https://doi.org/10.1016/j.cbpc.2022.109343) could provide a solid foundation for discussing modern methods and their application in current research.
-
The mention of ethical concerns is commendable, but the topic could be expanded to include specific guidelines or best practices for ethical handling during experiments with fish.
-
The conclusion of the introduction could be more impactful by highlighting how the study fills gaps in the literature and its specific scientific contribution. This will help guide the reader toward the relevance of the research.
Suggestions for Improvement
-
Restructure the introduction to connect the importance of blood flow in human organisms to its relevance in fish studies. For instance, emphasize how fish serve as important models for understanding cardiovascular mechanisms due to the transparency of their blood vessels and other experimental advantages.
-
Formulating a good hypothesis would be helpful in supporting the conclusions.
-
Eliminate repetitions and simplify some sentences to improve clarity. For example, instead of "Blood flow serves as the most important biologic parameter that can dictate the function of the heart," use "Blood flow is a critical parameter that directly influences heart function."
-
Expand the ethical section, detailing practices that ensure the welfare of fish during experiments. Cite international guidelines such as the Directive 2010/63/EU on the protection of animals used for scientific purposes.
-
Conclude the introduction with a statement that clearly indicates the objectives and relevance of the study, such as:
"This study aims to evaluate blood flow measurement techniques in fish, contributing to the development of non-invasive methodologies that can enhance cardiovascular research and improve ethical standards in aquatic animal studies." -
The methodology section presents a comprehensive and detailed description of the experimental protocols; however, there are several areas that could benefit from clarification and improvement to enhance scientific rigor and reproducibility. Below are constructive critiques and suggestions for improvement:
- While the acclimatization period is commendable, the methodology lacks detailed information on how the elimination of external factors was ensured. For instance, were there specific tests or observations conducted to confirm that the fish were acclimatized?
- Include details on the monitoring process during acclimatization, such as behavioral or physiological parameters assessed, to confirm the elimination of external stressors. Additionally, mention whether the water quality parameters (e.g., ammonia, nitrate levels) were monitored during this period
- The preparation of chemical solutions is described, but it would be beneficial to specify whether controls were tested for potential solvent effects, as acetone and DMSO can influence biological processes.
- Include details on the use of negative controls to account for any effects caused by the solvents themselves. For example, outline whether zebrafish were exposed to DMSO or acetone alone to rule out confounding effects.
- While the temperature gradient study is well-described, it is unclear if the exposure duration (24 hours) is sufficient to stabilize physiological responses to the new temperature. Additionally, the rationale for selecting specific tail vessel positions is not elaborated.
- Explain the rationale for the chosen exposure duration and how it ensures physiological stabilization. Provide more justification for selecting specific tail vessel positions, perhaps based on prior studies or pilot experiments.
- The inclusion of power analysis is a strength; however, the description of the statistical approach could be expanded for better transparency. For example, what level of confidence and margin of error were used?
- Specify the confidence interval and margin of error applied in the calculations. Include the results of the power analysis (e.g., calculated sample size) and discuss any limitations related to the sample size.
- The use of high-speed videography is innovative, but the description of the anesthesia process could be more detailed, particularly regarding potential stress or physiological effects on blood flow.
- Provide details on the duration of exposure to MS-222, how the anesthetic concentration was determined, and any measures taken to ensure it did not alter blood flow velocity. Include information on recovery procedures post-anesthesia.
- The method adaptation from Santoso et al. [18] is well-noted, but more details are needed to understand the applicability of the adult zebrafish settings to larvae.
- Explain the validation process for confirming that adult zebrafish blood cell size and flow parameters are comparable to those of larvae. Include a brief description of any limitations this adaptation might introduce.
- The exposure conditions are described, but the choice of compound concentrations lacks justification. Additionally, the description of the experimental design could benefit from more specificity regarding replication and statistical analysis.
- Provide a rationale for selecting the concentrations of carbofuran and copper oxide, such as references to prior studies or toxicity thresholds. Elaborate on the statistical methods used to analyze triplicate results and ensure reproducibility.
- While the study mentions adherence to IACUC guidelines, it does not elaborate on specific ethical measures taken to minimize stress or harm to the fish.
- Include more details on ethical protocols, such as measures to reduce stress during handling, exposure, and post-experiment care.
- discussion
-
The study provides valuable insights into cost-effective methods for measuring blood flow velocity in adult zebrafish, with potential applications in drug testing and environmental studies. However, several aspects of the discussion could be improved to enhance the clarity, rigor, and contextual depth of the conclusions:
-
While the method's simplicity and cost-effectiveness are emphasized, the discussion could benefit from a more explicit comparison to existing techniques in terms of accuracy, limitations, and scalability. This would position the study within the broader context of zebrafish cardiovascular research.
-
The limitations of the method are acknowledged, but the implications of these challenges—such as pigment interference in tiger barb or overlapping vessels in convict cichlid—could be further explored. Suggesting potential solutions, like using imaging filters to address pigmentation issues or advanced algorithms to separate vessel directions, would make the study more forward-looking.
-
The validation with pharmacological agents (FEN and PON) is well-discussed, but the conclusion regarding the absence of significant effects from CUO and CAR would benefit from a more detailed exploration. For example, the possibility of extending exposure duration or using higher doses to capture delayed or subtle effects could be proposed.
-
Given that CUO and CAR are environmental contaminants, it would be impactful to discuss how the findings contribute to understanding real-world exposure scenarios, such as chronic low-dose exposure in natural ecosystems.
-
While the method is shown to work for some species with similar fin-tail structures, the discussion could address its applicability to a broader range of aquatic organisms or its potential adaptation for specific research needs, such as developmental studies or species with unique cardiovascular anatomies.
-
The dependence on video quality and clarity is a key limitation. Highlighting recent advancements in imaging technology, such as machine learning for video enhancement or particle tracking, could inspire future improvements to the method.
-
Author Response
Comments and Suggestions for Authors
- The introduction addresses two main topics: the importance of blood flow for cardiovascular physiology and methods for its measurement in fish. However, the transition between these topics could be clearer. The relationship between blood flow in humans and its measurement in fish is not explicitly stated, which may hinder the reader's understanding of the overall relevance. It is recommended to reorganize the introduction to better integrate these themes.
Thank you for the suggestion. The authors also agree that there is some missing part that connect those topics. Thus the authors already add another paragraph in the introduction part the discuss about the importance of studying cardiovascular system in animal
- Many sections describe blood flow in general terms, with statements like "blood flow serves as the most important biologic parameter". While true, such claims would be more impactful if supported by specific examples or data. This approach helps justify the study's relevance and its contribution to the field.
Thank you for the suggestion. The authors agree that more specific example should be added as the reference to justify the study. Thus, the manuscript has been revised according to the suggestion.
- References are used to support assertions, but some information could be enriched with more recent or specific citations, especially regarding measurement techniques in fish. The suggested article (https://doi.org/10.1016/j.cbpc.2022.109343) could provide a solid foundation for discussing modern methods and their application in current research.
Thank you for the suggestion. The revised manuscript has been updated with more recent finding according to the reviewer suggestion.
- The mention of ethical concerns is commendable, but the topic could be expanded to include specific guidelines or best practices for ethical handling during experiments with fish.
Thank you for the recommendation. The authors agree that the addition of several reference regarding the ethical issue need to be addressed in the study to boast the advantages of the current study. However, the authors think that making a whole paragraph regarding the ethics will make the study out of topic. Thus, the manuscript has been adjusted according to the reviewer suggestion.
- The conclusion of the introduction could be more impactful by highlighting how the study fills gaps in the literature and its specific scientific contribution. This will help guide the reader toward the relevance of the research.
Thank you for the suggestion. The authors also agree that the conclusion of introduction lack some impact especially in the contribution to relevant research. Thus, the introduction section has been revised following the reviewer suggestion.
Suggestions for Improvement
- Restructure the introduction to connect the importance of blood flow in human organisms to its relevance in fish studies. For instance, emphasize how fish serve as important models for understanding cardiovascular mechanisms due to the transparency of their blood vessels and other experimental advantages.
Thank you for the suggestion. The authors also agree that there is some missing part that connect those topics. Thus the authors already add another paragraph in the introduction part the discuss about the importance of studying cardiovascular system in animal.
- Formulating a good hypothesis would be helpful in supporting the conclusions.
Thank you for the suggestion. The supporting hypothesis has been added in the last paragraph of introduction section.
- Eliminate repetitions and simplify some sentences to improve clarity. For example, instead of "Blood flow serves as the most important biologic parameter that can dictate the function of the heart," use "Blood flow is a critical parameter that directly influences heart function."
Thank you for the suggestion. The authors has been eliminate the possible repetition and simplifying the paragraph in the updated manuscript.
- Expand the ethical section, detailing practices that ensure the welfare of fish during experiments. Cite international guidelines such as the Directive 2010/63/EU on the protection of animals used for scientific purposes.
Thank you for the wonderful suggestion. The authors also agree that the addition of the mentioned ethical section will give a better understanding on how the method will be carried on. Thus the updated manuscript has been revised according to reviewer suggestion
- Conclude the introduction with a statement that clearly indicates the objectives and relevance of the study, such as:
"This study aims to evaluate blood flow measurement techniques in fish, contributing to the development of non-invasive methodologies that can enhance cardiovascular research and improve ethical standards in aquatic animal studies."
Thank you for the constructive suggestion. The author also agree that the introduction does not explicitly indicate the objective of the study. Thus, the authors already change the last part of the introduction section according to the reviewer suggestion.
- The methodology section presents a comprehensive and detailed description of the experimental protocols; however, there are several areas that could benefit from clarification and improvement to enhance scientific rigor and reproducibility. Below are constructive critiques and suggestions for improvement:
- While the acclimatization period is commendable, the methodology lacks detailed information on how the elimination of external factors was ensured. For instance, were there specific tests or observations conducted to confirm that the fish were acclimatized?
- Include details on the monitoring process during acclimatization, such as behavioral or physiological parameters assessed, to confirm the elimination of external stressors. Additionally, mention whether the water quality parameters (e.g., ammonia, nitrate levels) were monitored during this period
Thank you for the suggestion. Although the authors did not perform specific test to make sure of the acclimatization process. The acclimation process done to make sure there is no remnant of parasite during transportation process and in general this process done in about 2-4 weeks (Westerfield, 2000). The external factor mentioned in the first sub-section of the material and method refer to the difference of growth condition from supplier and our facility like the pH, nitrate, feeding time, etc. However, the author agree that this might confuse the reader, thus the updated manuscript has been change to accommodate this issue. Furthermore, the addition of other water quality parameter also added in the updated manuscript.
Westerfield, M. (2000). The Zebrafish Book: A Guide for the Laboratory Use of Zebrafish (Danio Rerio). University of Oregon Press. https://books.google.com.tw/books?id=Iy8PngEACAAJ
- The preparation of chemical solutions is described, but it would be beneficial to specify whether controls were tested for potential solvent effects, as acetone and DMSO can influence biological processes.
- Include details on the use of negative controls to account for any effects caused by the solvents themselves. For example, outline whether zebrafish were exposed to DMSO or acetone alone to rule out confounding effects.
Thank you for the suggestion. As mentioned in the section 2.3, the acetone of 0.01% and DMSO of 0.1% was used as the positive solvent control. The selection of concentration was based on the highest concentration of both solvent used to dissolve the compound used in the current study. The authors did not do the comparison of solvent with water as negative control to minimize the usage of fish in accordance with 3R policy. Futhermore as previous study mentioned, acetone up to 0.1% didn’t cause significant alteration in adult zebrafish behavior while 0.1 % of DMSO might alter the behavior after 10 days of continuous exposure (Audira et al., 2020) and in larvae stage, up to 0.1% concentration of acetone and DMSO done minimal change to the heart rate (Hallare et al., 2006). As the nature of exposure was different which only done once through oral galvage 24 hours prior to recording, the concentration that circulated in the system will be less and shorter compared to the one done in the previous study. However the author also agree that in the result section all the solvent control was stated as control which might confusing the reader. Thus, the author already do some change in the updated manuscript to address this issue.
Audira, G., Siregar, P., Chen, J.-R., Lai, Y.-H., Huang, J.-C., & Hsiao, C.-D. (2020). Systematical exploration of the common solvent toxicity at whole organism level by behavioral phenomics in adult zebrafish. Environmental Pollution, 266, 115239.
Hallare, A., Nagel, K., Köhler, H.-R., & Triebskorn, R. (2006). Comparative embryotoxicity and proteotoxicity of three carrier solvents to zebrafish (Danio rerio) embryos. Ecotoxicology and environmental safety, 63(3), 378-388.
- While the temperature gradient study is well-described, it is unclear if the exposure duration (24 hours) is sufficient to stabilize physiological responses to the new temperature. Additionally, the rationale for selecting specific tail vessel positions is not elaborated.
- Explain the rationale for the chosen exposure duration and how it ensures physiological stabilization. Provide more justification for selecting specific tail vessel positions, perhaps based on prior studies or pilot experiments.
Thank you for the detailed comment. Although a previous study stated that the zebrafish can tolerate up to 1°C change per day (Åsheim et al., 2020), the aim of the temperature gradient experiment was to show that the change in ambience holding temperature especially in low temperature can change the blood flow velocity in zebrafish rather than showing the overall physiological response of the fish at difference ambience temperature. The 24 hours waiting time was to make sure that the fish was depraved of possible spike of cardiovascular activity due to shocked during handling process when moving to and sudden change of temperature which might increase their cardiovascular activity.
The authors select the middle position in both base and tip of caudal fin due to the those position was parallel to the heart position of zebrafish. The authors agree that this part lack justification and might confused the reader. Thus the manuscript has been updated according to the reviewer suggestion.
Åsheim, E. R., Andreassen, A. H., Morgan, R., & Jutfelt, F. (2020). Rapid-warming tolerance correlates with tolerance to slow warming but not growth at non-optimal temperatures in zebrafish. Journal of Experimental Biology, 223(23), jeb229195.
- The inclusion of power analysis is a strength; however, the description of the statistical approach could be expanded for better transparency. For example, what level of confidence and margin of error were used?
- Specify the confidence interval and margin of error applied in the calculations. Include the results of the power analysis (e.g., calculated sample size) and discuss any limitations related to the sample size.
Thank you for the reminder. The level of confidence was set to 80% confidence while the margin of error was set to 5% as stated in the first section of result together with the result of power analysis ,which used to select the best position to do recording. The author agree that this might confused the reader. Thus the selection of confidence rate and error has been moved to material and method section in the revised manuscript.
- The use of high-speed videography is innovative, but the description of the anesthesia process could be more detailed, particularly regarding potential stress or physiological effects on blood flow.
- Provide details on the duration of exposure to MS-222, how the anesthetic concentration was determined, and any measures taken to ensure it did not alter blood flow velocity. Include information on recovery procedures post-anesthesia.
Thank you for the suggestion. Although the author cannot synchronize the duration of MS-222 immersion time due to difference of the time needed to reach total anesthesia stage for every fish species is different, the author agree that specific stage of anesthesia should be explained. Thus the stage of anesthesia has been added in the manuscript together with the post-anesthesia procedure in the sub-section 5 of material and method section.
- The method adaptation from Santoso et al. [18] is well-noted, but more details are needed to understand the applicability of the adult zebrafish settings to larvae.
- Explain the validation process for confirming that adult zebrafish blood cell size and flow parameters are comparable to those of larvae. Include a brief description of any limitations this adaptation might introduce.
Thank you for the suggestion. The authors agree that this need to be briefly address to justify the method usage, Thus some image showing the comparison between adult and zebrafish larvae has been added as supplementary figure.
- The exposure conditions are described, but the choice of compound concentrations lacks justification. Additionally, the description of the experimental design could benefit from more specificity regarding replication and statistical analysis.
- Provide a rationale for selecting the concentrations of carbofuran and copper oxide, such as references to prior studies or toxicity thresholds. Elaborate on the statistical methods used to analyze triplicate results and ensure reproducibility.
Thank you for the wonderful suggestion. The concentration selection of carbofuran and copper oxide was based on previous study in larvae that show significant alteration after exposure. The author agree this might cause confusion. Thus, the rationale of compound concentration selection has been added for all compound use in the study. Furthermore, although the statistical tool used already mentioned in each figure in result section, some confusion might be happened due to this issue. Thus the statistical methodology has been elaborated more in the updated manuscript.
- While the study mentions adherence to IACUC guidelines, it does not elaborate on specific ethical measures taken to minimize stress or harm to the fish.
- Include more details on ethical protocols, such as measures to reduce stress during handling, exposure, and post-experiment care.
Thank you for the concern. As the mentioned guideline didn’t specifically address the handling of fish but animal in general, the authors cannot elaborate more regarding the guideline. However the authors agree that detail handling during several experiment especially the experiment that need anesthesia need to be address in the manuscript together with the post-experimental care. Thus,(Liu et al., 2013) the updated manuscript has been updated according to the reviewer suggestion.
- discussion
- The study provides valuable insights into cost-effective methods for measuring blood flow velocity in adult zebrafish, with potential applications in drug testing and environmental studies. However, several aspects of the discussion could be improved to enhance the clarity, rigor, and contextual depth of the conclusions:
- While the method's simplicity and cost-effectiveness are emphasized, the discussion could benefit from a more explicit comparison to existing techniques in terms of accuracy, limitations, and scalability. This would position the study within the broader context of zebrafish cardiovascular research.
Thank you for the comment. The limitation of the current study has been addressed in last paragraph of the discussion section. However, as the author knowledge this is the first study that do the blood flow calculation in adult tail of zebrafish. Thus to comparing with other method regarding their accuracy will be quite difficult for us to do. Regarding the scalability, the authors already add the test in another fish like medaka, black tetra, and other several bigger fish to test their versatility. In that consideration the authors believes that thus method can be used in other laboratory that do different animal than zebrafish.
- The limitations of the method are acknowledged, but the implications of these challenges—such as pigment interference in tiger barb or overlapping vessels in convict cichlid—could be further explored. Suggesting potential solutions, like using imaging filters to address pigmentation issues or advanced algorithms to separate vessel directions, would make the study more forward-looking.
Thank you for the suggestion. The authors agree with reviewer that this topic can be address more to enhance the manuscript quality. Thus the updated manuscript has been change according to the reviewer suggestion.
- The validation with pharmacological agents (FEN and PON) is well-discussed, but the conclusion regarding the absence of significant effects from CUO and CAR would benefit from a more detailed exploration. For example, the possibility of extending exposure duration or using higher doses to capture delayed or subtle effects could be proposed.
Thank you for the comment. Although not explicitly, the author already suggest that the incubation of 24 hours might be not enough for the CuO and CAR to takes effect which might be caused by the mechanism of toxicity and absorption rate. Furthermore, the testing in using CuO and CAR was based on the previous study in larvae, in the current study, the author just want to replicate the same setting as the one used in larvae. The authors agree that addition of more experiment will cause the manuscript out of topic especially since this study is focused more on finding new method than doing toxicity test. However, the authors agree that some separate research regarding this issue will be interesting for future study that focused more on toxicity testing.
- Given that CUO and CAR are environmental contaminants, it would be impactful to discuss how the findings contribute to understanding real-world exposure scenarios, such as chronic low-dose exposure in natural ecosystems.
Thank you for the comment. Previous study regarding the chronic exposure of CUO has been discussed in the discussion section at paragraph. However back ground chronic study was add for carbofuran. Thus the manuscript has been revised according to the reviewer suggestion
- While the method is shown to work for some species with similar fin-tail structures, the discussion could address its applicability to a broader range of aquatic organisms or its potential adaptation for specific research needs, such as developmental studies or species with unique cardiovascular anatomies.
Thank you for the wonderful suggestion. The authors also agree that this method have a lot of potential and applicable to a broad range of research field. Thus, the discussion part has been updated according to the reviewer suggestion
- The dependence on video quality and clarity is a key limitation. Highlighting recent advancements in imaging technology, such as machine learning for video enhancement or particle tracking, could inspire future improvements to the method.
Thank for the suggestion. The author also agree that this issue can give future solution for the current limitation found in the study. Thus the updated manuscript has been revised according to the reviewer suggestion
Åsheim, E. R., Andreassen, A. H., Morgan, R., & Jutfelt, F. (2020). Rapid-warming tolerance correlates with tolerance to slow warming but not growth at non-optimal temperatures in zebrafish. Journal of Experimental Biology, 223(23), jeb229195.
Audira, G., Siregar, P., Chen, J.-R., Lai, Y.-H., Huang, J.-C., & Hsiao, C.-D. (2020). Systematical exploration of the common solvent toxicity at whole organism level by behavioral phenomics in adult zebrafish. Environmental Pollution, 266, 115239.
Hallare, A., Nagel, K., Köhler, H.-R., & Triebskorn, R. (2006). Comparative embryotoxicity and proteotoxicity of three carrier solvents to zebrafish (Danio rerio) embryos. Ecotoxicology and environmental safety, 63(3), 378-388.
Liu, T.-Y., Lee, P.-Y., Huang, C.-C., Sun, L., & Shung, K. K. (2013). A study of the adult zebrafish ventricular function by retrospective Doppler-gated ultrahigh-frame-rate echocardiography. IEEE transactions on ultrasonics, ferroelectrics, and frequency control, 60(9), 1827-1837.
Westerfield, M. (2000). The Zebrafish Book: A Guide for the Laboratory Use of Zebrafish (Danio Rerio). University of Oregon Press. https://books.google.com.tw/books?id=Iy8PngEACAAJ
Round 2
Reviewer 1 Report
Comments and Suggestions for Authors
I am satisfied of the corrections.